# Effect of Tempering Temperature on Microstructure and Mechanical Properties of Cr-Ni-Mo-V Rotor Steel

**DOI:** 10.3390/ma18030555

**Published:** 2025-01-26

**Authors:** Chao Zhao, Xinyi Zhang, Xiaojie Liang, Guowang Song, Bin Wang, Liqiang Guo, Pengjun Zhang, Shuguang Zhang

**Affiliations:** 1School of Mechanical and Electrical Engineering, North University of China, Taiyuan 030051, China; 20220012@nuc.edu.cn (C.Z.); sz202201045@st.nuc.edu.cn (X.Z.); 2Taiyuan Heavy Industry Co., Ltd., Taiyuan 030051, China; liangxiaojienancy@126.com (X.L.); 13834130137@139.com (G.S.); 3First Machinery Group Firmaco Shanxinorth Machine Buliding Co., Ltd., Taiyuan 030051, China; 13934619666@139.com (B.W.); guoliqiang163.com@163.com (L.G.); 4College of Materials Science and Engineering, Hohai University, Nanjing 211100, China

**Keywords:** tempering temperature, Cr-Ni-Mo-V rotor steel, mechanical properties, carbide

## Abstract

In this paper, we investigated the effects of the matrix and precipitates in Cr-Ni-Mo-V rotor steel on its mechanical properties after water quenching and tempering (450–700 °C). The results indicate that the microstructure and mechanical properties of the steel can be significantly adjusted by changing the tempering temperature. An excellent combination of tensile strength (1028.608 MPa) and elongation (19%) was obtained upon tempering at 650 °C. This is attributed to the martensite lath with a high dislocation density, solid solution strengthening and the strengthening effect of spherical Mo_2_C and VC particles. At a tempering temperature of 550 °C, the precipitation and development of rod-shaped Fe_3_Mo_3_C resulted in a considerable drop in strength. At 650 °C, the dissolution of Fe_3_Mo_3_C and dispersion precipitation of Mo_2_C and VC led to a large rise in strength. At 700 °C, the coarsening of Mo_2_C and VC, together with the recrystallization of the martensite lath, resulted in a loss in strength. Meanwhile, as the tempering temperature was increased from 450 °C to 700 °C, the tensile fracture characteristics of Cr-Ni-Mo-V rotor steel gradually changed from cleavage fractures to dimple fractures.

## 1. Introduction

Large rotors are an important part of much industrial equipment, and are widely used in power generation, aerospace, metallurgy, the chemical industry, mining and other fields [1,2]. Since rotors experience enormous stress and centrifugal force during high-speed rotation, rotor steel must have a sufficiently high yield strength and tensile strength to prevent plastic deformation and fracture. The medium-carbon, low-alloy Cr-Ni-Mo-V steel has excellent mechanical properties and good workability and is often used in various large rotors [3]. Its excellent mechanical properties are mainly due to the strengthening of the alloying elements and the complex microstructure formed during heat treatment. Normalizing, quenching and tempering, as a common heat treatment process for Cr-Ni-Mo-V steel, can effectively adjust the microstructure and performance of the material by controlling the process temperature and holding time. Specifically, the tempering process constitutes the final stage of the heat treatment process, and its impact on the mechanical properties is of paramount importance [4].

During the tempering process, the microstructure of Cr-Ni-Mo-V rotor steel undergoes several transformation processes. In particular, the martensite formed after quenching decomposes with increasing tempering temperatures and forms tempered martensite. In addition, the characteristics of the carbides are one of the important factors affecting the fatigue properties [5], impact toughness [6] and corrosion wear properties [7] of Cr-Ni-Mo-V rotor steel. The carbide transformation in the microstructure is closely related to the tempering temperature. Moreover, the characteristics (location, type, quantity and volume) of the carbide have a significant influence on crack initiation and propagation [8]. Tempering at high temperatures usually promotes the formation of large carbides, while tempering at low temperatures contributes to the refinement of these carbides, which in turn affects the mechanical properties of the material [9]. Although a large number of publications have discussed the effects of tempering temperature on the properties of Cr-Ni-Mo-V rotor steel [10,11,12,13,14], there is still a lack of systematic research on the subtle mechanism between tempering temperature and microstructure. Currently, the normalizing temperature for Cr-Ni-Mo-V steel during the heat treatment process is set at 870 °C, with the quenching temperature at 840 °C and the tempering temperature at 600 °C. This steel exhibits a tensile strength of 950 MPa and an elongation of 18%. Given the small increase in the Mo and V alloy components within the Cr-Ni-Mo-V steel addressed in this research, changes to the heat treatment method are required. However, increasing the quenching temperature might cause grain development, which reduces mechanical characteristics. Thus, it is critical to investigate the microstructure and property development of Cr-Ni-Mo-V steel under various tempering procedures.

At different tempering temperatures, the change in the matrix and the precipitated phase of the steel has a considerable influence on its properties. In general, the strength of the steel is high and the plasticity is low at low tempering temperatures. The higher the tempering temperature, the lower the strength of the steel, but the higher the toughness [15]. Therefore, the choice of appropriate tempering temperature to compensate for the strong plasticity of Cr-Ni-Mo-V rotor steel has become a key problem to be solved. However, the mechanism of the influence of tempering temperature on the microstructure and properties of Cr-Ni-Mo-V rotor steel is still controversial. In existing studies, the mechanical properties of Cr-Ni-Mo-V rotor steel generally decrease with the increase in tempering temperature, mainly because the dislocation density of the matrix decreases and the coarsening of carbide precipitation leads to a weakening of the dislocation strengthening and solution strengthening effects [16,17,18]. However, some studies have found that the secondary strengthening effect of Cr-Ni-Mo-V rotor steel usually occurs with an increase in tempering temperature, and is mainly influenced by the precipitation and strengthening of the carbide [19,20,21]. Therefore, it is of great theoretical significance and technical benefit to investigate the influence mechanism of microstructure evolution on the properties of Cr-Ni-Mo-V rotor steel during tempering.

In this paper, the microstructure and mechanical properties of Cr-Ni-Mo-V rotor steel at different tempering temperatures were systematically studied by means of tensile tests, scanning electron microscopy (SEM), transmission electron microscopy (TEM) and X-ray diffraction (XRD). The influence mechanism of tempering temperature on the properties of Cr-Ni-Mo-V rotor steel was revealed. The aim of this research is to provide a theoretical basis for optimizing the heat treatment process of Cr-Ni-Mo-V rotor steel and to promote its wide application in the field of large rotors.

## 2. Materials and Methods

In the paper, the Cr-Ni-Mo-V rotor steel specimens were taken from the annealed rotor after forging, and its specific chemical composition is shown in Table 1. The specimens were placed into the muffle furnace (the thermocouple is located in the center of the back wall of the muffle furnace, and the temperature difference in each area of the furnace is ±5 °C) for normalizing, quenching and tempering. Figure 1a,b demonstrate the heat treatment method and CCT curves for the Cr-Ni-Mo-V rotor steel. Figure 1c–e illustrate the microstructures following annealing, normalizing and quenching. Following the completion of the heat treatment, the bar material tempered at different temperatures was processed into tensile specimens, as shown in Figure 2.

The microstructure was analyzed with the JSM-7001F SEM (JEOL Ltd., Tokyo, Japan), Tecnai F30 TEM (Thermo Fisher Scientific, Hillsboro, OR, USA) and Smartlab 9 kW XRD (Rigaku Corporation, Tokyo, Japan). The SEM specimen was mechanically ground, polished and then etched in a 4% (wt.%) Nital. A TEM specimen with a thickness of 60 μm was hand-ground with sandpaper and then punched into a disk with a diameter of 3 mm. Then, TEM samples were prepared by twin-jet electropolishing using an electrolyte composed of 90% alcohol and 10% perchloric acid at 25 °C and 40 V. The Cr-Ni-Mo-V rotor steel samples were cut with a surface area of 10 × 10 × 10 mm, sandpapered and mechanically polished, and XRD analysis was performed. The scanning angle 2θ was 10–100°, and the scanning speed was 4°/min. The XRD results were calibrated using MDI Jade5.0 software to determine the distance between the crystal faces of the sample. Three tensile specimens were tested at each tempering temperature, and their average values were taken for analysis. The tensile properties of the samples were tested by Z600E 600N electronic (ZwickRoell, Ulm, Germany) tensile testing machine, and then the fracture morphology was observed by SEM.

## 3. Results and Discussion

### 3.1. Mechanical Properties After Quenching and Tempering

Figure 3a,b show the engineering stress–strain curves and tensile property tendencies of Cr-Ni-Mo-V rotor steel at room temperature under different tempering conditions. The data on the tensile properties are shown in Table 2. Figure 3b illustrates that when the tempering temperature rises from 450 °C to 700 °C, the tensile and yield strength fall, then rise again, but the elongation and reduction in area steadily increase. This is primarily due to martensitic recovery and the recrystallization of Cr-Ni-Mo-V rotor steel, as well as carbide dissolution and precipitation. Tensile and yield strength decrease from 997.091 MPa and 800.75 MPa to 828.902 MPa and 688.655 MPa, respectively, between 450 °C and 550 °C, whereas elongation and area reduction increase from 13% and 61% to 18.2% and 69%. At 650 °C, the tensile strength and yield strength increase to 1028.608 MPa and 860.271 MPa, respectively, while elongation and decrease in area increase by 19% and 71.88%. At 700 °C, tensile and yield strength decrease to 754.004 MPa and 644.205 MPa, respectively, while elongation and reduction in area increase to 22.4% and 76.57%. In conclusion, the comprehensive mechanical properties of Cr-Ni-Mo-V rotor steel achieved their optimal state when tempered at 650 °C, with a tensile strength of 1028.608 MPa and elongation of 19%, which are higher than the strength and elongation of traditional rotor steel. This may be connected to the recovery and recrystallization of the martensitic lath in the microstructure of Cr-Ni-Mo-V rotor steel, as well as the evolution of the type and volume of carbides [22,23].

### 3.2. Microstructure Evolution During Tempering

Figure 1c–e show the microstructure of Cr-Ni-Mo-V rotor steel under the annealing, normalizing and quenching processes before tempering. According to the analysis of Figure 1b, the matrix and carbides show different characteristics after annealing, normalizing and quenching due to different cooling rates. Figure 4a–d show the SEM microstructure of Cr-Ni-Mo-V rotor steel at various tempering temperatures, whereas Figure 4e–h show the carbide energy spectrum. At 450 °C, the microstructure of Cr-Ni-Mo-V rotor steel consists of martensite lath, rod-shaped Fe_3_Mo_3_C, and a minor quantity of spherical VC, as shown in Figure 4a,e. Figure 4b,f show that at 550 °C, a substantial quantity of rod-shaped Fe_3_Mo_3_C in Cr-Ni-Mo-V rotor steel is spread at the grain boundary, whereas a little amount of VC and Mo_2_C is dispersed within the grain. Compared to 450 °C, the amount of Fe_3_Mo_3_C increases and the diameter increases from 200 nm to 500 nm at 550 °C, and it was concluded that a large amount of Fe_3_Mo_3_C precipitated and grew. The content of Fe_3_Mo_3_C dramatically drops at 650 °C, as shown in Figure 4c,g, while a considerable quantity of VC and Mo_2_C disperses. Figure 4d,h show that at 700 °C the matrix’s Fe_3_Mo_3_C content decreases, VC and Mo_2_C coarsen and the precipitation-promoting impact on Cr-Ni-Mo-V rotor steel weakens.

In conclusion, a large amount of Fe_3_Mo_3_C and a small amount of VC and Mo_2_C precipitated from the matrix in the tempering range of 450 °C to 550 °C, suggesting that the weakening of the solution’s strength may be related to the strength drop in this temperature range [24]. The amount of VC and Mo_2_C in the matrix increases while the amount of Fe_3_Mo_3_C decreases in the tempering range of 550 °C to 650 °C, suggesting that precipitation hardening may be the reason of the strength increase. The quantities of VC and Mo_2_C grow while the amount of Fe_3_Mo_3_C in the matrix falls in the tempering range of 650 °C to 700 °C, demonstrating that the weakening of precipitation strengthening is associated with the strength decline [25].

Figure 5 depicts the variations in the lattice constant of alpha-Fe, the matrix diffraction intensity and the dislocation density of Cr-Ni-Mo-V rotor steel at various tempering temperatures. Figure 5a demonstrates that the spacing of the matrix’s crystal face reduces when tempered at 450–550 °C, increases when tempered at 550–650 °C and decreases when tempered at 650–700 °C. This is consistent with the shifting trend of strength with tempering temperature (Figure 3b). The change in the crystal face spacing of the matrix also reflects the solid solution and precipitation law of alloying elements, which is also confirmed by the precipitation and resolution of Fe_3_Mo_3_C carbide in the matrix and the precipitation and coarsening of VC and Mo_2_C carbide in Figure 4. Figure 5b,c show the matrix diffraction strength and the dislocation density which is determined using the half-peak width of the diffraction peak at different tempering temperatures. With the tempering temperature increasing, the diffraction intensity and dislocation density decreased gradually due to the gradual recovery and recrystallization of martensite lath.

Figure 6 shows the TEM microstructure of Cr-Ni-Mo-V rotor steel at different tempering temperatures. From Figure 6a, under the tempering condition of 450 °C, the martensitic lattice microstructure in Cr-Ni-Mo-V rotor steel is still the main microstructure form, with a large number of rod-shaped carbides distributed in the grain boundaries and a small number of spherical carbides distributed in the martensitic lattice and boundary, which corresponds to Figure 4a. In addition, dislocation proliferation and entanglement occurred near the carbides, and the low recovery of the martensite lamella led to a high dislocation density [26]. The carbide’s restriction of dislocation movement and the high dislocation density in the martensitic lath promoted the maintenance of the material’s high strength [27]. From Figure 6b, when the tempering temperature increases to 550 °C, a large amount of rod-shaped Fe_3_Mo_3_C precipitates at the grain boundary, and a small amount of spherical VC and Mo_2_C in the grains leads to a weakening of the strengthening effect of the solution, in which the hindering effect of dislocation is weakened by the increase in the volume of Fe_3_Mo_3_C and the dislocation density decreases due to the higher recovery degree of martensitic lath. Therefore, the strength of Cr-Ni-Mo-V rotor steel decreased as a result of solution strengthening, dislocation strengthening and precipitation strengthening. Figure 6c shows that cellular structures were formed during tempering at 650 °C due to the local recrystallization of martensitic laths, which leads to a decrease in dislocation density. In particular, the dissolution of rod-shaped Fe_3_Mo_3_C led to an increased strength increase in the matrix in solid solution, while the diffusion of spherical VC and Mo_2_C increased the hindrance of dislocations [28]. The strength of Cr-Ni-Mo-V rotor steel increased with the enhancement of solution strength and precipitation strength. It can be seen from Figure 6d that when the tempering temperature increases to 700 °C, almost all martensitic lath recrystallizes to form cellular structures, which leads to a further reduction in dislocation density. At the same time, the high tempering temperature led to excessive precipitation and the growth of VC, Mo_2_C and Fe_3_Mo_3_C carbides [8]. Therefore, strengthening by dislocation, solution and precipitation were weakened simultaneously, resulting in a significant decrease in material strength [29]. The above microstructural changes can clearly explain the evolution of the mechanical properties of Cr-Ni-Mo-V rotor steel at different tempering temperatures (Figure 3). This paper clarified the transition law of steel carbide with increasing tempering temperature. In the future, the transition mechanism of carbides should be further studied to provide a theoretical basis for further improvements of properties.

### 3.3. Fracture Morphology

The tensile specimen fracture of Cr-Ni-Mo-V rotor steel at different temperatures is shown in Figure 7. As shown in Figure 7a, the microcracks at the tempering temperature of 450 °C mainly propagate through the grain boundaries or within the grains without significant plastic deformation regions. The tensile fracture is mainly a brittle fracture, which has the characteristics of transgranular fractures with flat or small crack extensions. As illustrated in Figure 7b, increasing the tempering temperature to 550 °C results in a brittle fracture surface with transgranular cracks and grain boundary fractures, as well as modest local plastic deformation of the dimpling structure. The tensile fracture shows a quasi-brittle fracture. Compared to tempering at 450 °C, tempering at 550 °C may produce more grain boundary fractures and minor dimple structures, indicating a little increase in plasticity but no major ductile fracture pattern. As shown in Figure 7c, when the tempering temperature increases to 650 °C, the fracture has an obvious dimple structure, and the dimple shape is deep but small, showing an obvious ductile fracture. When the tempering temperature increases to 700 °C, the tensile fracture characteristics still remain ductile, and the fracture surface still has a dimple structure, but the dimple shape becomes shallow and large. To summarize, when the tempering temperature increases from 450 °C to 700 °C, the tensile fracture characteristic of Cr-Ni-Mo-V rotor steel gradually change from cleavage fractures to dimple fractures.

From Figure 4a and Figure 6a, it can be seen that when tempering at 450 °C, a large amount of Fe_3_Mo_3_C at the grain boundary easily causes stress concentrations, which lead to crack initiation and propagation, and a low elongation and reduction in area lead to insufficient plastic deformation [30]. As a result, microcracks form at the grain boundary and eventually fracture intergranularly or transgranularly. It can be seen from Figure 4b and Figure 6b that during tempering at 550 °C, a large amount of Fe_3_Mo_3_C gradually grows at the grain boundaries and martensitic lath boundaries, the dislocation density within the grains decreases, and the precipitation of alloying elements weakens the effect of solution strength and dislocation strength [31]. As a result, microcracks can occur simultaneously at the grain boundary and inside the grain, leading to a quasi-crack fracture. As shown in Figure 4c and Figure 6c, Fe_3_Mo_3_C redissolution at the grain boundary and martensite lath enhances the solid solution strengthening of the matrix, and spherical VC and Mo_2_C precipitate in large amounts at the martensite lath boundary. Thereby, microcrack can easily occur in the grain, and the high strain and reduction in area make the microstructure susceptible to local plastic deformation. The tensile specimens show pitting fractures [32]. Figure 4d and Figure 6d show that during tempering at 700 °C, spherical VC and Mo_2_C coarsen along the martensite lath border, leading to crack formation and propagation [33]. Compared with tempering at 650 °C, the dimpled shape of tensile fractures at 700 °C is flatter and larger.

## 4. Conclusions

(1)The microstructure of the Cr-Ni-Mo-V rotor steel in this work is mainly tempered martensite. With an increasing in tempering temperature, the tensile strength and yield strength first decrease, then increase and then decrease again, while elongation and section shrinkage gradually increase, with 650 °C having the best comprehensive mechanical properties.(2)The strength of Cr-Ni-Mo-V rotor steel is mainly influenced by dislocation strengthening in the matrix, the solid solution strengthening of the carbon and alloying elements and the precipitation strengthening of carbides. The tensile strength and plasticity are the best after tempering at 650 °C, which is mainly due to the comprehensive effect of Fe_3_Mo_3_C dissolution leading to the solid solution strengthening of the matrix and the precipitation strengthening of VC and Mo_2_C.(3)When the tempering temperature increases from 450 °C to 700 °C, the tensile fracture properties of Cr-Ni-Mo-V rotor steel gradually change from cleavage fracture to pitting fracture, which is mainly affected by the dissolution, precipitation and growth of Fe_3_Mo_3_C, VC and Mo_2_C in the structure.(4)In order to give theoretical support for further optimization of Cr-Ni-Mo-V rotor steel characteristics and heat treatment processes, future research should investigate the mechanism of carbide transition from Fe_3_Mo_3_C to Mo_2_C and VC.

## Figures and Tables

**Figure 1 materials-18-00555-f001:**
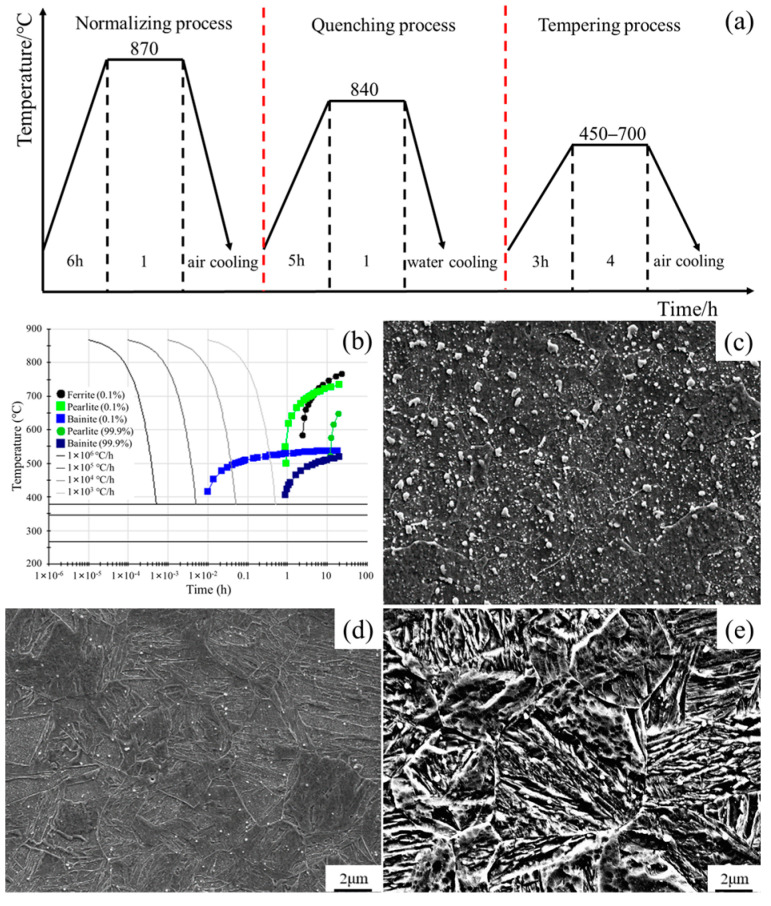
Heat treatment procedure (**a**), CCT curve (**b**), annealed structures (**c**), normalized structures (**d**) and quenched structures (**e**) of Cr-Ni-Mo-V rotor steel.

**Figure 2 materials-18-00555-f002:**
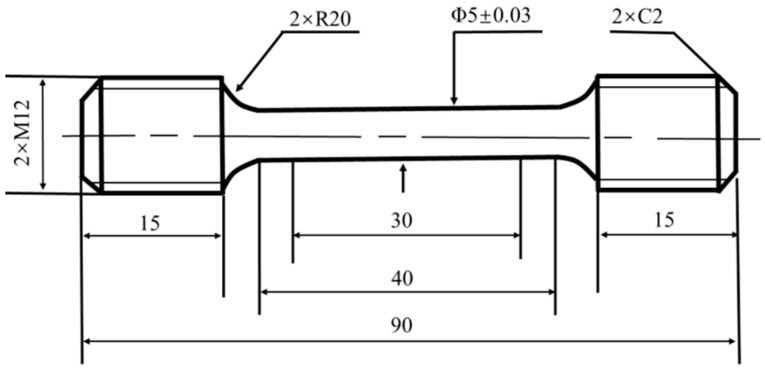
Tensile sample of Cr-Ni-Mo-V rotor steel.

**Figure 3 materials-18-00555-f003:**
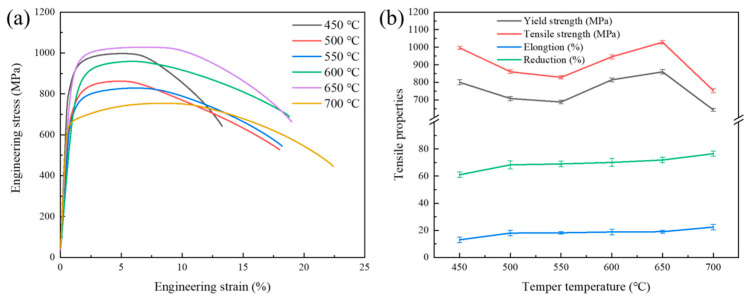
Engineering stress–strain curves (**a**) and tensile properties (**b**) of Cr-Ni-Mo-V rotor steel at various tempering temperatures.

**Figure 4 materials-18-00555-f004:**
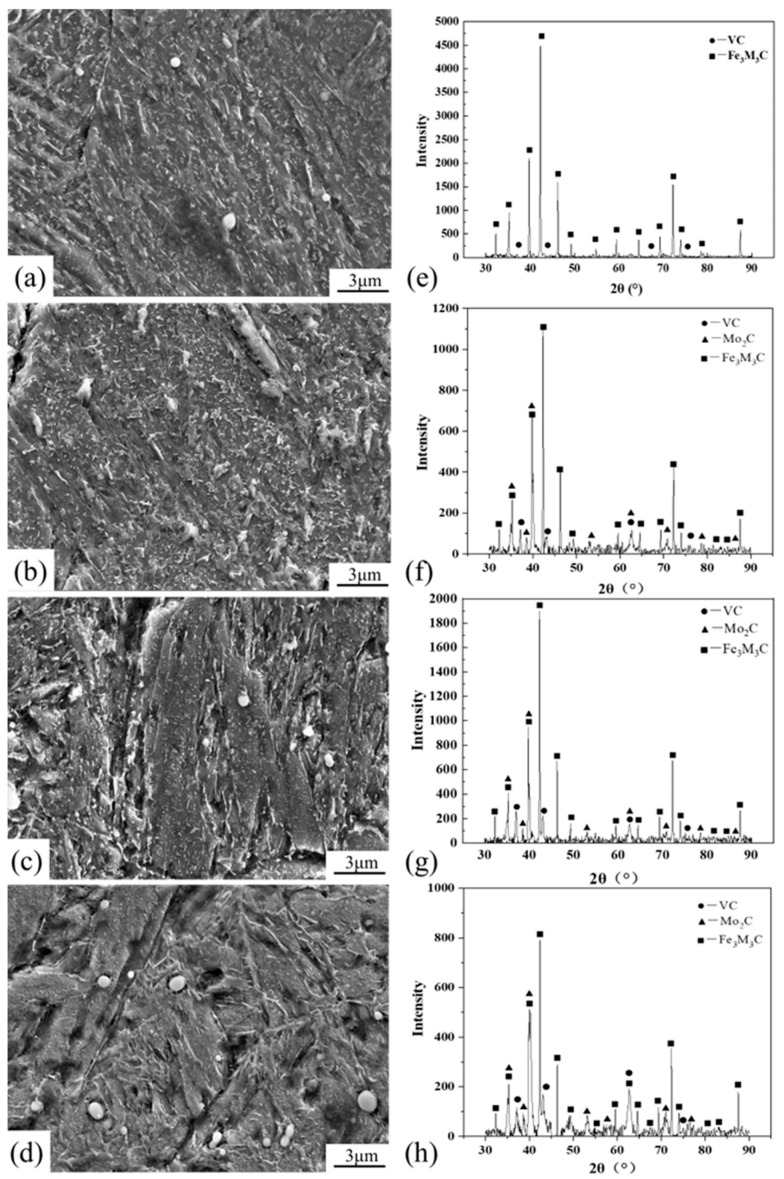
SEM microstructure (**left**) and energy spectrum (**right**) of Cr-Ni-Mo-V rotor steel at various tempering temperatures: (**a**) and (**e**) 450 °C; (**b**) and (**f**) 550 °C; (**c**) and (**g**) 650 °C; (**d**) and (**h**) 700 °C.

**Figure 5 materials-18-00555-f005:**
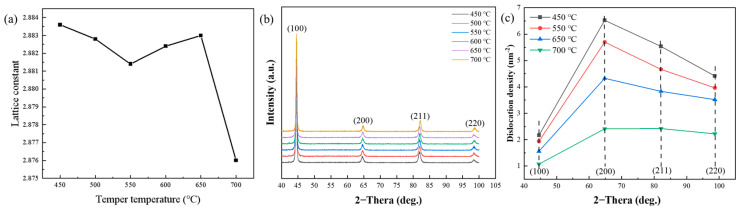
Lattice constant of alpha-Fe (**a**), matrix diffraction intensity (**b**) and dislocation density (**c**) of Cr-Ni-Mo-V rotor steel at different tempering temperatures.

**Figure 6 materials-18-00555-f006:**
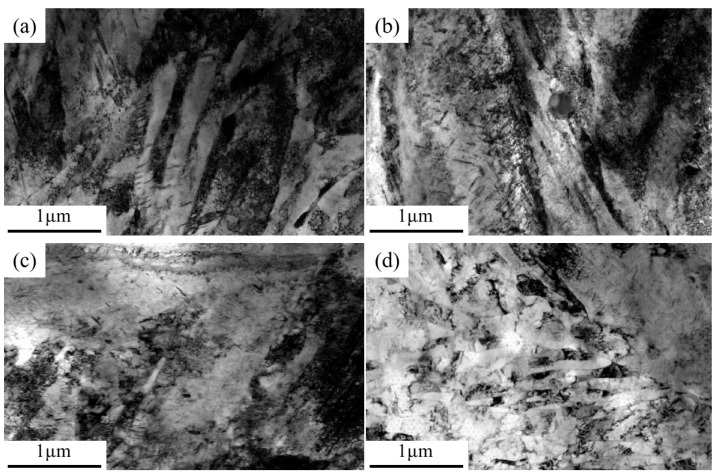
TEM microstructure of Cr-Ni-Mo-V rotor steel: (**a**) 450 °C; (**b**) 550 °C; (**c**) 650 °C; (**d**) 700 °C.

**Figure 7 materials-18-00555-f007:**
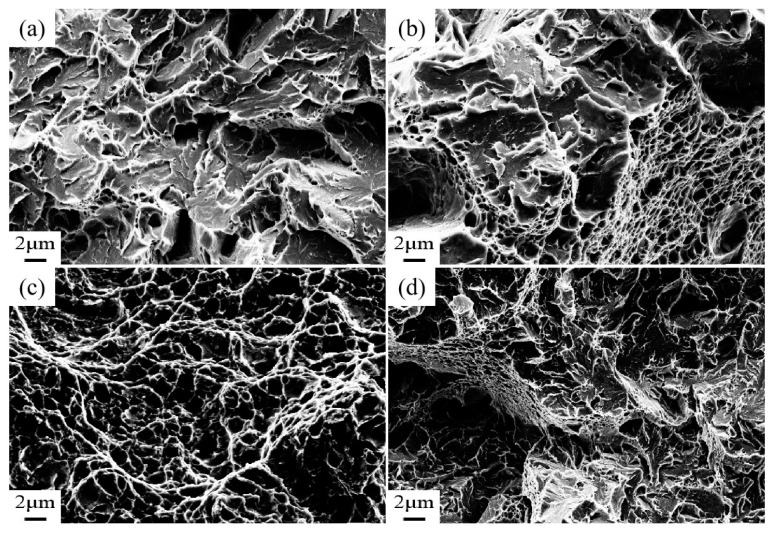
Tensile fracture morphology of Cr-Ni-Mo-V rotor steel: (**a**) 450 °C; (**b**) 550 °C; (**c**) 650 °C; (**d**) 700 °C.

**Table 1 materials-18-00555-t001:** Main chemical composition of Cr-Ni-Mo-V rotor steel (wt.%).

Element	C	Cr	Mo	W	Ni	V
Standard	0.25–0.28	2.5–2.8	1.6–1.9	0.2–0.6	0.5–0.8	0.2–0.5
Experimental	0.27	2.68	1.75	0.40	0.68	0.42

**Table 2 materials-18-00555-t002:** Tensile mechanical properties of Cr-Ni-Mo-V rotor steel at various tempering temperatures.

Temperature (°C)	Yield Strength (MPa)	Tensile Strength (MPa)	Elongation (%)	Reduction (%)
450	800.75	997.091	13	61
500	708.469	861.016	18	68.34
550	688.665	828.902	18.2	69
600	814.953	945.651	18.8	70.11
650	860.271	1028.608	19	71.88
700	644.205	754.004	22.4	76.57

## Data Availability

Data are contained within the article.

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
