# Peer review of "Effect of Tempering Temperature on Microstructure and Mechanical Properties of Cr-Ni-Mo-V Rotor Steel"

_materials, 2025, doi:10.3390/ma18030555_

Round 1
Reviewer 1 Report
Comments and Suggestions for Authors
The manuscript must be reformatted in accordance with MDPI requirements.
It is necessary to refer to the fatigue strength of Cr–Ni–Mo–V Rotor Steel, its impact strength, resistance to corrosion, abrasion and erosive wear and the effect of tempering temperature on them.
It is necessary to refer to the role and form of carbides and the phenomenon of delaminate crack in the fracture surface depending on tempering temperature.
In 'Conclusions' it is necessary to provide proposed directions for future research.
Generally, it is better to place the legend outside the graph area than on the graph.
Figure 3 - it is better to place the table separately than on the graph.
Figure 4 - in the figure description, explain what is placed on the left and what on the right side of the figures.
The bibliography needs to be slightly updated - a lot is happening in the analyzed area.
Comments on the Quality of English LanguageChecking English is recommended.
Reviewer 2 Report
Comments and Suggestions for Authors
The paper explores heat treatments at maximum temperatures between 450 to 700 ° C in a Cr–Ni–Mo–V steel. Characterization techniques such as X-ray, SEM, and the tensile test were used to evaluate the influence of the study variable. Results show a cyclic variation of the yield and tensile strength as the temper temperature increased, while the elongation was not affected in the range of temperature tested. I have some minor comments to improve the paper presentation:
–it is necessary to structure the abstract to be attractive to any reader. Please, consider rewriting and include the Introduction and purpose, then the research problem, Methodology, results, and finally the main conclusions.
–In the introduction, The process regularly used is not explained, please introduce the tempering temperature and tensile resistance of the current Cr–Ni–Mo–V steels. Also, the improvements that could be generated by your study should be discussed in the results.
–Please, include the time scale in Figure 1. It is not evident the time of each stage of the heat treatment.
- Figure 3 – correct "tensile properties"
Reviewer 3 Report
Comments and Suggestions for Authors
MANUSCRIPT REVIEW
Manuscript title: Effect of Tempering Temperature on Microstructure and Mechanical Properties of Cr–Ni–Mo–V Rotor Steel
The objective of this study is the influence of tempering temperature (in the range of 450-700 ℃) on the microstructure and mechanical properties of Cr–Ni–Mo–V rotor steel for large rotors. The following characterization methods were used: SEM (scanning electron microscope), TEM (transmission electron microscope), XRD (X-ray diffraction), and tensile test. After different tempering temperatures, the change in the matrix and the precipitated phase of the steel has a strong influence on its properties. With the increase of temperature, tensile strength and yield strength of Cr–Ni–Mo–V rotor steel decreased, then increased and then decreased again, and the best comprehensive mechanical properties were obtained at 650 °C. The mechanical properties (tensile strength and plasticity) are the best after tempering at 650 °C, which is mainly due to the comprehensive effect of Fe3Mo3C dissolution leading to solid solution strengthening of the matrix and precipitation strengthening of VC and Mo2C.
Reviewer’s Comments:
1. Line 94, it is necessary to emphasize whether it is a vol.% or wt.% solution of nitro-alcohol.
2. Line 99,-20 â—¦C”, please use an adequate symbol.
3. Line 101: Please define the type and model of the XRD device.
4. Figure 4. The proposal is to separately group images from SEM and energy spectrum analysis. Please see the instructions for authors for arranging figures.
5. Line 188: "Degree of martensitic recovery of Lat.”. Lat.?
6. Please use subscript for labeling chemical formulas.
7. Prepare a list of references according to the template. The style of references (list) is not uniform and not according to the requirements of the journal.
18.12. 2024.

Reviewer 4 Report
Comments and Suggestions for Authors
In this manuscript the mechanical properties of Cr–Ni–Mo–V rotor steel have been discussed through TEM, SEM, XRD and uniaxial tensile test. From my opinion of view the manuscript could be accepted after minor corrections.
Some particular comments
The authors claim; “The aim of this study is to systematically analyze the evolution of the microstructure and mechanical properties of Cr–Ni–Mo–V rotor steel at different tempering temperatures and to reveal the mechanism of the influence of tempering temperature on the properties of Cr–Ni–Mo–V rotor…..The aim of this research is to provide a theoretical basis for optimizing the heat treatment process of Cr–Ni–Mo–V rotor steel and to promote its wide application in the field of large rotors….The authors must provide a main objective only,
On page 4 line 142, The author affirm that compared to 450 ℃, the content and volume of Fe3Mo3C in the tempered microstructure increased at 550 ℃. The authors must provide the Fe3Mo3C volume in absolute value (%).
The authors are speculative, due to the mechanical properties depend of the dislocation density; the authors must provide HRTEM images than show dislocations density on the treated samples (tempering temperature).
Reviewer 5 Report
Comments and Suggestions for Authors
This is good research work on heat treatment of Cr–Ni–Mo–V Rotor Steel. However, there are some minor issues that need to be addressed, as mentioned below:
· There are some English related issues such as
o In Abstract, on line 22 its not lat martensite, it is lath martensite.
o Please take care of subscripts such as Fe3Mo3C must be written as Fe3Mo3C
o Therefore, the change in strength is mainly influenced by the balance between the strengthening of the solid solution, the strengthening of the precipitation and the strengthening of the dislocation in the matrix. Please avoid repetition.
Materials and Methods:
· In this test, in line 83 refers to?
· 4% nitroalcohol solution is wrong, I think you used 4% Nital, please confirm and correct it.
· TEM sample preparation method is not correctly explained, it seems you polished the sample using electrolytic as well Focus ion beam. Please explain it properly as in 96 to 99 it seems that you are mixing up TEM sample preparation methods.
Results and Discussion:
· In lines 108 to 111, I found below paragraph, I am not sure if these were guidelines in template or it something coming from generative AI. Please confirm,
‘’Authors should discuss the results and how they can be interpreted from the perspective of previous studies and of the working hypotheses. The findings and their implications should be discussed in the broadest context possible. Future research directions may also be highlighted.’’
· As this paper is focused on heat treatment, thus I would recommend before going to mechanical properties, it is necessary to show microstructures (Optical or SEM) of as received, forged, normalised, quenched and after various tempering.
· Effect of heat treatment on mechanical properties is discussed using figure 3 however it seems that only one test was performed as statistical information such as standard deviation or standard error is not given,
· As there is difference of 50 ℃ in tempering temperatures, please explain how much temperature variation as was there in furnace thermocouples and how much temperature was consistent in different areas of furnace chamber. Even in small furnaces ± 10-15℃ is quite common and in large furnaces this variation can go up to 50℃. Please explain thermocouple locations and temperature variation in furnace.
· Please provide full tensile test stress strain curves as for base material and for all heat-treated conditions as tabulated data is not enough to discuss mechanical behaviour and effect of tempering. Stress strain curves provide much more details about yielding, strain hardening, flow of material, UTS and fracture etc.
· Please include data for all of characterisation for base material, as received, forged, normalised, quenched condition as well and then discuss effect of and after various tempering temperatures. CCT or TTT curves must be included to explain the evolution of different phases.
Please review these points and make the necessary corrections to ensure clarity and consistency throughout the paper.
Comments on the Quality of English Language
Please see above
Round 2
Reviewer 5 Report
Comments and Suggestions for Authors
This manuscript is qutie improved as in this revised version authors have addressed comments.